# Structural basis of promiscuous substrate transport by Organic Cation Transporter 1

Yi C. Zeng [1,2] ✉, Meghna Sobti[1,2], Ada Quinn [3], Nicola J. Smith [4], Simon H. J. Brown [5], Jamie I. Vandenberg [2,6], Renae M. Ryan [7], Megan L. O'Mara [3] & Alastair G. Stewart [1,2] ✉

Organic Cation Transporter 1 (OCT1) plays a crucial role in hepatic metabolism by mediating the uptake of a range of metabolites and drugs. Genetic variations can alter the efficacy and safety of compounds transported by OCT1, such as those used for cardiovascular, oncological, and psychological indications. Despite its importance in drug pharmacokinetics, the substrate selectivity and underlying structural mechanisms of OCT1 remain poorly understood. Here, we present cryo-EM structures of full-length human OCT1 in the inward-open conformation, both ligand-free and drug-bound, indicating the basis for its broad substrate recognition. Comparison of our structures with those of outward-open OCTs provides molecular insight into the alternating access mechanism of OCTs. We observe that hydrophobic gates stabilize the inward-facing conformation, whereas charge neutralization in the binding pocket facilitates the release of cationic substrates. These findings provide a framework for understanding the structural basis of the promiscuity of drug binding and substrate translocation in OCT1.

The Solute Carrier 22 (SLC22) family of membrane transporters facilitates the movement of endogenous and exogenous compounds across cell membranes[1–3]. These transporters are part of the larger major facilitator superfamily (MFS) that share a common transmembrane architecture and utilize an alternating access mechanism based on cycling between outward-facing and inward-facing states[4,5]. However, unlike many known MFS transporters that are specific solute carriers[5,6], SLC22 transporters are non-selective carriers of a range of small molecule cations and anions and are critically important in the absorption and elimination of nutrients, metabolites, drugs, and toxins in a range of organs[1,3,7,8].

Organic Cation Transporter 1 (OCT1), a member of this family primarily localized on the basolateral membrane of hepatocytes, is involved in the uptake of cationic molecules for hepatic metabolism and excretion[9,10]. OCT1 is known to transport many endogenous metabolites and dietary nutrients including thiamine (vitamin B1)[3,11], a cofactor for several enzymes in the catabolic pathway. Unlike the specific thiamine transporters (ThTr1/2, *SLC19A2/3*), OCT1 exhibits a lower affinity and higher transport velocity for thiamine, likely acting to facilitate the uptake of high doses of this nutrient during oral absorption[11]. Genetic variants that result in a reduction of OCT1 transport of thiamine have been linked to several cardiovascular and metabolic diseases and could also be influenced by drug co-administration[12–14].

OCT1 is also involved in the pharmacokinetics of many small molecule therapeutics, including metformin, fenoterol, and diltiazem[3].

[1]Molecular, Structural and Computational Biology Division, The Victor Chang Cardiac Research Institute, Darlinghurst, NSW, Australia. [2]School of Clinical Medicine, Faculty of Medicine and Health, UNSW Sydney, Sydney, NSW, Australia. [3]Australian Institute of Bioengineering and Nanotechnology, University of Queensland, Brisbane, QLD, Australia. [4]School of Biomedical Sciences, Faculty of Medicine & Health, UNSW Sydney, Kensington, NSW, Australia. [5]School of Chemistry and Molecular Bioscience, Molecular Horizons, and Australian Research Council Centre for Cryo-electron Microscopy of Membrane Proteins, University of Wollongong, Wollongong, NSW, Australia. [6]Molecular Cardiology and Biophysics Division, The Victor Chang Cardiac Research Institute, Darlinghurst, NSW, Australia. [7]School of Medical Sciences, Faculty of Medicine and Health, University of Sydney, Sydney, NSW, Australia. ✉e-mail: y.zeng@victorchang.edu.au; a.stewart@victorchang.edu.au

Metformin, a front-line anti-diabetic drug, is a well-known substrate of OCT1. Genetic polymorphisms of OCT1 are known to reduce the effectiveness of metformin by lowering absorption and causing gastrointestinal intolerance to metformin[15–18]. Fenoterol, a $\beta_2$ adrenergic agonist widely used as an anti-asthmatic and labor suppressant, is also a substrate for OCT1. Hereditary loss of function of OCT1 can lead to accumulating concentrations of fenoterol in the circulation, in turn increasing the risk of adverse cardiovascular events[19]. Transport of metformin by OCT1 can also be impacted through competitive inhibition by other drugs, resulting in an altered toxicity risk[20,21]. Reduction of uptake into the liver through OCT1 inhibition by commonly prescribed drugs has been reported to alter the metabolism and clearance of cardiovascular therapeutics, leading to potentially fatal interactions[22]. Additionally, anti-arrhythmic drugs such as diltiazem, a calcium channel blocker, exhibiting plasma serum concentrations above the $IC_{50}$ of OCT1 after administration and could similarly contribute to cardiovascular toxicity[23].

There are broad implications for drug discovery and clinical practice in evaluating and understanding how new and existing compounds interact with OCT1[3,21,24–26]. Although recent structural work on human OCT3 and an engineered consensus sequence of OCT1 and OCT2 ($OCT1_{CS}$ and $OCT2_{CS}$, respectively) has aided the understanding of the mechanism of inhibition of OCTs[27,28], a precise appreciation of how diverse scaffolds with varying biochemical profiles are transported by or inhibit OCTs remains elusive. Using cryo-electron microscopy (cryo-EM), we examine a panel of chemically and functionally diverse substrates complexed to human wild-type full-length OCT1. We observe substrate-free and substrate-bound OCT1 in the inward-open conformation and comparison of these structures with that of outward-open OCTs gives insight into the molecular mechanism of cation transport versus inhibition of OCTs. These structures identify key interactions between different substrates and OCT1, providing a structural context for understanding how new and existing pharmaceuticals could be transported in the liver or blocked by competitive drug binding.

## Results
### Conformational changes in human OCT1 from outward- to inward-facing states
We solved cryo-EM structures of human OCT1 in the absence of ligands (OCT1-apo, 2.9 Å resolution), and to compounds of varying affinities (Supplementary Fig. 1, Supplementary Table 1): OCT1 bound to thiamine/vitamin B1 (OCT1-THA, 3.1 Å resolution), OCT1 bound to metformin (OCT1-MTF, 3.5 Å resolution), OCT1 bound to fenoterol (OCT1-FNT, 3.2 Å resolution), and OCT1 bound to diltiazem (OCT1-DTZ, 3.3 Å resolution) (Fig. 1, Supplementary Figs. 2–7, Supplementary Table 2). The overall architecture of OCT1 in our cryo-EM structures resembles the classical MFS transmembrane arrangement of pseudo-symmetric 6 + 6 transmembrane helical (TM) bundles corresponding to the N-terminal (TMs 1–6) and C-terminal (TMs 7–12) lobes (Fig. 1b)[5,6]. MFS-fold transporters use an alternating access mechanism based on cycling between outward- and inward-facing conformations through an intermediate occluded transition state during the translocation of substrates[5,6]. The 6 + 6 TM bundles of our OCT1 cryo-EM structures are arranged in an inward-open conformation with the central substrate cavity accessible to the cytoplasm. The structures also show a large extracellular domain (ECD) between TMs 1 and 2 unique to the SLC22 family. A series of intracellular helices (ICH), corresponding to the sequence between the two lobes and the N-terminus, were not resolved in our structures (Fig. 1a, b). To further probe the protein structure, replicate 500 ns molecular dynamics (MD) simulations ($n = 3$) were performed on the model obtained here for OCT1-apo (residues K19-P283 and P332-T516, Supplementary Table 3). During these simulations, the TM domain of OCT1 did not undergo significant conformational changes, with backbone root

mean square deviation (RMSD) of $2.1 \pm 0.3$ Å over the combined 1500 ns ($n = 3 \times 500$ ns) of production simulations, suggesting that the inward-open confirmation is stable on these timescales in silico (Supplementary Table 4).

When compared to the outward-open conformation of human OCT3 and $OCT1_{CS}$[27,28], our inward-open structure of human OCT1 showed distinct conformational shifts in the C-terminal lobe (Fig. 2a, b, Supplementary Movie 1). When aligned on the N-terminal lobe, TM7 was rotated inwards by 29°, TM9 by 25°, TM10 by 31° and TM12 by 26° (Supplementary Fig. 8a). TM8 was rotated only in the intracellular half by 20° through a helix break by $P388_{OCT1}$ (Supplementary Fig. 8a). A fenestration to the outer leaflet in outward-open OCT3 caused by a helical break at $G480/G481_{OCT3}$ ($G476/G477_{OCT1}$) on TM11 was closed in by 36° in our inward-open structure and instead was rotated by 26° towards the inner leaflet, opening another potential fenestration (Supplementary Fig. 8b). It has previously been demonstrated in rat OCT1 that mutation of $G478_{rat}$ ($G477_{human}$) to serine or cysteine reduces the flexibility of TM11 to be able to transition between inward and outward conformations[29]. In comparison to OCT structures, the glycine-proline motif of TM11 appears to act as a hinge allowing the upper segment of TM11 to undergo conformational shifts to fully open the binding site in the outward-open state as well as to then occlude the extracellular gate when changing to inward-open (Supplementary Fig. 8b).

Within these larger conformational changes, we observed changes in the interactions between the two lobes in the intracellular gate of OCT1. Salt-bridges formed between $E232_{OCT3}$ (TM4) to $R466_{OCT3}$ (TM11) and $R239_{OCT3}$ (TM5) to $E459_{OCT3}$ (TM10) in the outward conformation were dissociated upon transition to the inward state (Fig. 2c, d). In our structures of inward-open OCT1, density for the entire ICH bundle was not observed, suggesting that it is highly mobile in the inward-open conformation of OCT1. Instead, we observed weak density of TM6 and part of ICH2 in the N-terminal lobe deflected outward into the membrane from P271 in OCT1-MTF and OCT1-DTZ (Supplementary Fig. 9a, b). In outward-open OCT3, the ICH bundle was well resolved and contacted the ascending loop of TM7[27]. Comparison of TM6 between OCT1-MTF and OCT3 suggests that the ICH bundle would move away from the ascending loop of TM7 when going from outward- to inward-facing, inhibiting this interaction and forming a fully inward-open state (Supplementary Fig. 9c, d). This gating mechanism is analogous to transitions observed in the glucose transporter family[30].

The extracellular gate of OCT1 was stabilized by a combination of polar and hydrophobic interactions between the N- and C-terminal lobes. TM2 formed new inter-bundle polar interactions with TMs 7 and 11 (Fig. 2e). In MD simulations the average distance between the polar atoms of the residue D149-R486 and N156-Q362 sidechains was $3.4 \pm 0.6$ Å (consistent with a salt bridge) and $4.7 \pm 1.1$ Å (consistent with a water bridge), respectively (Supplementary Fig. 10d). In inward-open OCT1-apo, the complete closure of the extracellular gate was further stabilized by the union of hydrophobic residues from both lobes, shielding these residues from bulk solvent (Fig. 2e, Supplementary Fig. 10b, c). Simulations of OCT1-apo revealed that residues V37, I365 and L366, which sit in the center of the hydrophobic gate, were within 3 Å of any solvent molecule for only 8.2%, 23.4% and 23.6% of the combined 1500 ns ($n = 3 \times 500$ ns) of simulation time, respectively (Supplementary Fig. 10d, Supplementary Table 6), suggesting these residues of the predicted hydrophobic gate were tightly packed, preventing extracellular solvent penetration between the N- and C- bundles. In the structure of outward-occluded $OCT2_{CS}$, a similar interaction between $Y37_{OCT2}$ and $Y362_{OCT2}$ ($C36_{OCT1}$ and $Y361_{OCT1}$) was observed to form a thin extracellular gate over the substrate[28]. Additional deep scanning mutagenesis studies of human OCT1 have also shown that a number of the residues involved in the extracellular gate observed in

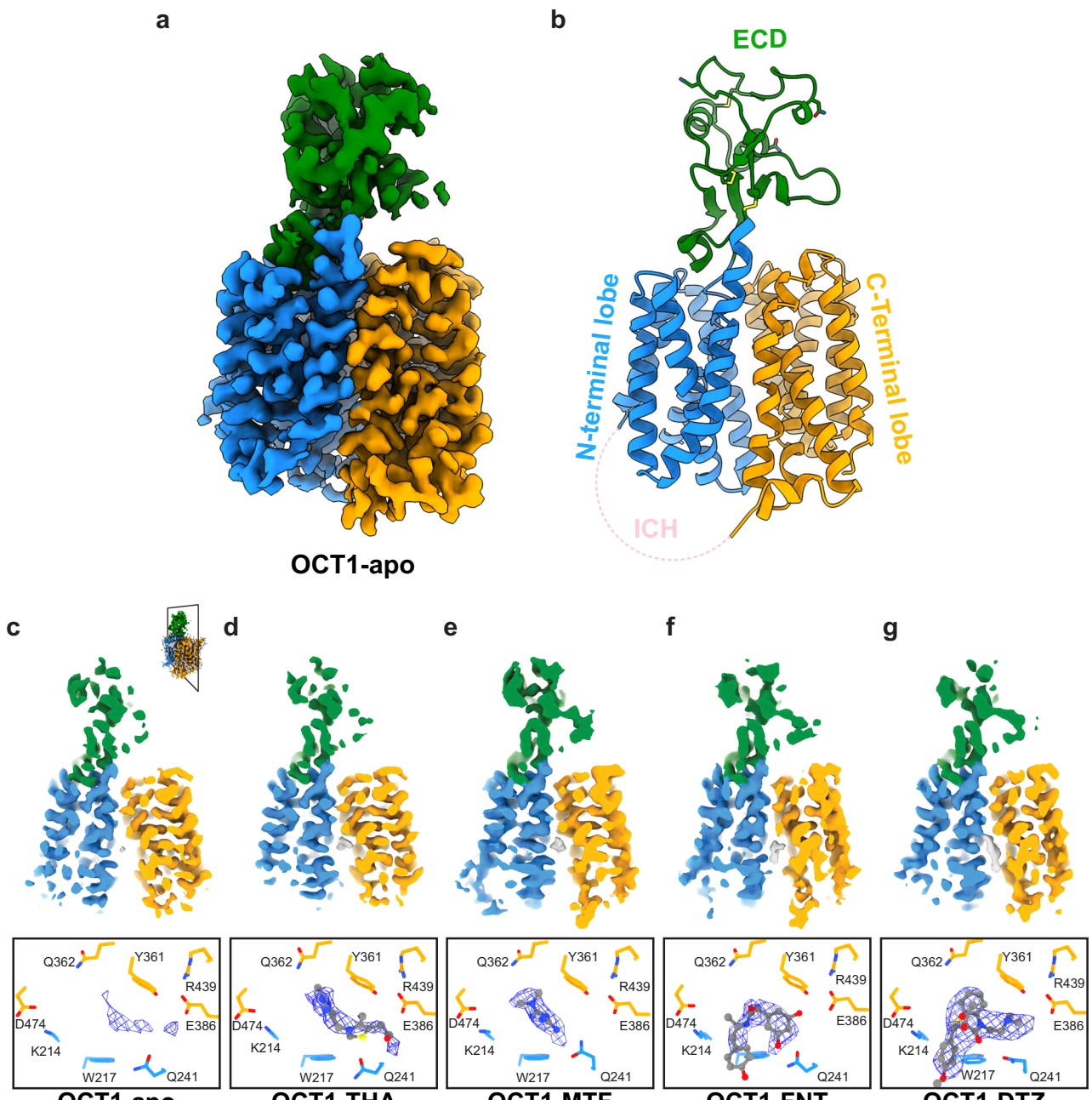

**Fig. 1 | Cryo-EM structures of Organic Cation Transporter 1 in the inward-open conformation.** OCT1 consists of four domains: N-terminal lobe colored in blue, C-terminal lobe colored in orange, extracellular domain (ECD) colored in green, and intracellular helices (ICH) colored in pink. **a** Cryo-EM map of OCT1-apo. **b** Backbone structure of OCT1-apo with domains labelled. **c**–**g** Cryo-EM maps of OCT1-apo and OCT1 after incubation with thiamine (OCT1-THA), metformin (OCT1-MTF), fenoterol (OCT1-FNT), and diltiazem (OCT1-DTZ). Top: cross-section of map perpendicular to the membrane, to show binding site (grey) relative to the structure (maps contoured to 9 σ). Bottom: close-up of binding site, with map around substrate shown as blue mesh (maps contoured to σ = 7.5). Further detailed views of binding site map density can be found in Supplementary Fig. 7.

our structures are also important in transport, as deleterious substitution (e.g. hydrophobic to polar and vice versa) of these residues results in reduced transport[31]. Interactions between the N- and C-terminal lobes of OCTs mimic mechanisms seen in the open to occluded transitions of other MFS transporters[32–34]. Comparison between the OCT3-outward and OCT1-inward structures demonstrated large twists of TMs 7, 10, and 11 during extracellular gate closure mediated through helix breaking glycines and prolines (Supplementary Fig. 8), consistent with other MFS transporters that exhibit these mechanisms of gating[35–37]. However, MFS transporters often rely on inter-bundle or protein-ligand polar interactions to

facilitate the gating transition[32,34,35], whereas the gating interaction between the N- and C-terminal lobes of OCTs appear to favor burying a large hydrophobic patch to stabilize this transition.

The entire ECD of OCT1-apo was resolved in our cryo-EM structure, albeit with regions of lower resolution towards the periphery (Fig. 3, Supplementary Fig. 2). We analyzed the flexibility of this domain further by first performing a focused local refinement on the transmembrane region of the protein, then performing 3D classification and 3D variability analysis in cryoSPARC[38] (Supplementary Fig. 11). We observed that, relative to the transmembrane region, the ECD still exhibited some rigid-body flexibility and torsion, however, this motion

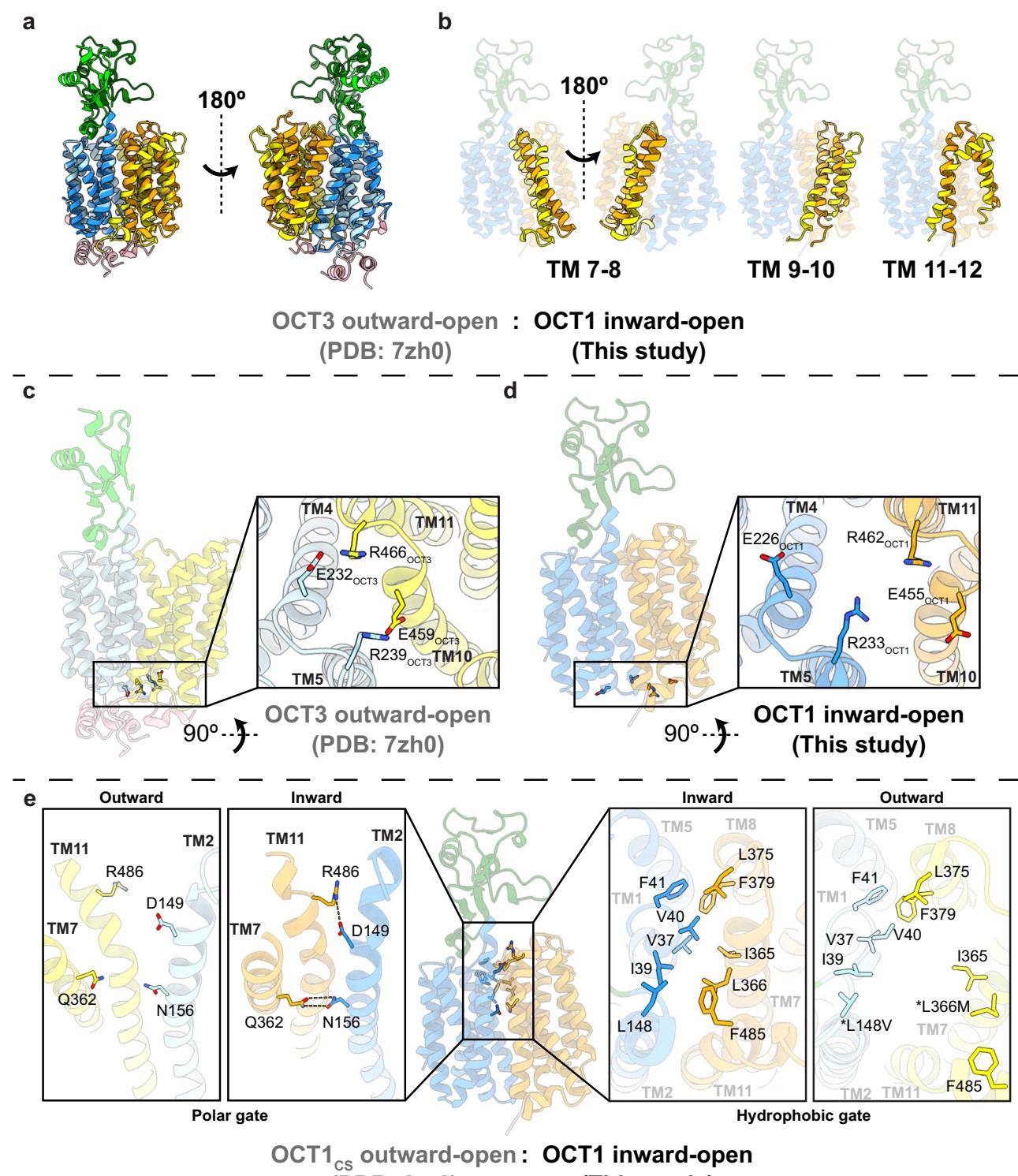

**Fig. 2 | Conformational changes between inward-open OCT1 and outward-open OCT3/OCT1CS involve gating residues. a**, **b** Superposition of the N-terminal lobes of OCT1-apo (darker colors) and OCT3 (PDB:7zh0[27], lighter colors). Relative to the N-terminal lobe, the C-terminal lobe pivots about the plane of the membrane and the ECD twists perpendicular to the membrane. Detailed comparison of each of the C-terminal lobe helices can be found in Supplementary Fig. 8. Location of residues involved in the intracellular gate between **c** OCT3, and **d** OCT1. **e** Location of extracellular gate residues involved in polar interactions (left) or hydrophobic contacts (right) in OCT1-apo and OCT1CS (PDB:8et6[28], lighter colors). Mutated residues in OCT1CS indicated by asterisks. Further views of extracellular gates in Supplementary Fig. 10a–c.

was quite subtle in the isolated 3D classes (Supplementary Fig. 11d). When compared with the partial ECD structure of outward-open OCT3[27], a relative tilt of 24° could be observed between these two domains (Supplementary Fig. 11e).

The core fold of the ECD is stabilized by three disulfide bonds: C50-C121, just after the anti-parallel beta sheet; C62-C102 towards the periphery of the ECD; and C88-C142 between the ECD and the start of TM2 (Fig. 3). Disulfide pairs C50-C121 and C88-C142 are conserved

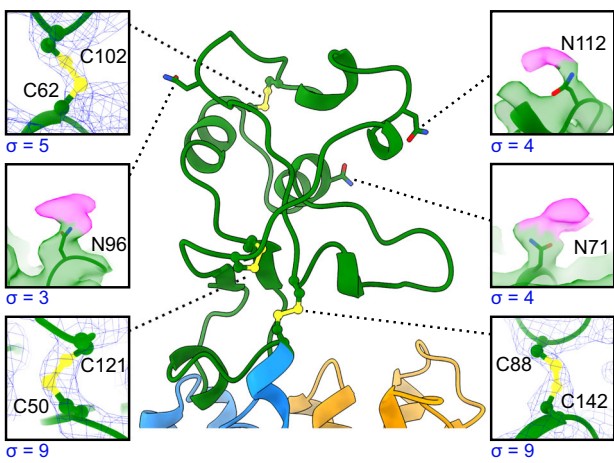

**Fig. 3 | The extracellular domain of OCT1 is stabilized by disulfide bonds and decorated with glycosylation sites.** Close-up of the ECD showing the cryo-EM map for the disulfides (yellow) and glycosylation sites with glycan density in pink. Map contour level labeled below each panel.

across SLC22 members, whereas C62-C102 is unique to OCTs (Supplementary Fig. 12). Our cryo-EM map of OCT1-apo in the inward-open state defines the entire ECD, whereas this region was only partially resolved in the previous outward-open OCT3 and OCT1$_{CS}$/OCT2$_{CS}$ structures, suggesting that the inward-open state restricts the flexibility of this domain. We also observed additional density that likely corresponded to the first glycan sugars on residues N71, N96, and N112 (Fig. 3). All three asparagine residues are on the surface of the ECD and sterically accessible, whereas N139, at the base of the ECD, is much more hindered and did not show corresponding glycan density. However, as the glycosylation-limiting HEK GnTI$^{-/-}$ cell line was used to express the protein used in this study, we are unable to explore the full extent to which glycosylation plays a functional role in this transporter.

### Interactions of OCT1 with low-affinity substrates

Thiamine and metformin are substrates of OCT1 with affinities in the millimolar range[11,39] (Supplementary Fig. 1, Supplementary Table 2). Because of the importance of reduced-function genetic variants and drug interactions of these substrates with each other and to other clinical drugs, we investigated the interactions of OCT1 with thiamine (vitamin B1, OCT1-THA) and metformin (OCT1-MTF).

The cryo-EM structures of OCT1-THA and OCT1-MTF in the inward conformation strongly resembled OCT1-apo (Fig. 1, Supplementary Figs. 2–4). The density corresponding to thiamine and metformin could only be observed at a lower threshold compared to the sidechain density (σ < 15 for sidechain, σ < 9 for ligand), likely owing to thiamine's low affinity for the transporter (Fig. 4a, c, Supplementary Fig. 7). However, the general contour of the ligands in the map could be established clearly and therefore we were able to model the probable orientation of these substrates in the binding pocket. We did also observe weak cryo-EM density in the binding site of OCT1-apo at low thresholds (σ < 9, Fig. 1c, Supplementary Fig. 7). Although we were unable to determine the identity of this material, we hypothesize this may be a buffer component or an endogenous cationic ligand, or the average of multiple different chemical species, consistent with the well-established ability of OCT1 to bind a variety of ligands. The general shape and volume of the ligand density in OCT1-apo was distinct from that observed with OCT1-THA or OCT1-MTF, where 10 mM of substrate had been incubated with the protein sample for cryo-EM analysis (the highest concentration we tested that still allowed for high resolution cryo-EM data collection and was above the reported K$_m$ of thiamine and metformin[11,39]).

Thiamine formed interactions with aromatic and polar residues of the inward-open binding pocket (Fig. 4a). Both heteroaromatic rings of thiamine were surrounded by predominantly aromatic residues, with the thiazolium ring forming cation-π interactions. Of note, the positive charge of the thiazolium ring was distant from either E386 or D474. E386 in the outward-open conformation was solvent-accessible and directly coordinated cations in other OCTs, whereas D474 appeared to be important in structural stability due to its consistent coordination to K214 and may influence the electrostatics of the binding pocket (Fig. 4f)[40,41]. The terminal alcohol of thiamine formed a hydrogen bond to Q447, potentially positioning the substrate towards the exit from the substrate cavity. However, as the density of thiamine was weaker than the protein, we independently characterized the interaction of thiamine and its tautomers with OCT1 by performing MD simulations (2 tautomers, each simulated for n = 3 replicates of 500 ns, Supplementary Fig. 13 and Supplementary Table 3, 6). In over 80% of the total 3000 ns of simulation time, the thiamine hydroxyl simultaneously donated a hydrogen bond to E386 and accepted a hydrogen bond from either Y361 or Q447 (Supplementary Fig. 13c). The thiazolium group was in contact with the aromatic sidechains of F244 and Y361 for more than 80% of the combined simulation time, whereas contact between thiazolium and the charged sidechains of either E386 or D474 was very infrequent (less than 5% of total simulation time, Supplementary Table 7). These observations are consistent with the interactions observed in the cryo-EM maps. However, throughout the simulations the pyrimidine ring adopted multiple orientations suggesting the conformation of the molecule is flexible in the binding site (Supplementary Fig. 13b). This observation is consistent with its low affinity as a substrate and the weaker density observed in the cryo-EM map.

Metformin showed unanticipated interactions between the biguanide and the substrate pocket (Fig. 4c, d). Metformin has a pKa of 11.5 and exists predominantly as a cation at physiological pH (7.4) and at the pH of the cryo-EM study (pH 8.0)[42]. Despite this, we were surprised to find that, like thiamine, metformin did not interact with the acidic residues E386 and D474 in the inward-open conformation (Fig. 4c). The density corresponding to metformin appeared to be coordinated by Q241 and so the compound was modelled with the desmethylated nitrogen forming putative polar interactions to this residue (Fig. 4c). However, in MD simulations of metformin and OCT1 (n = 3 × 500 ns replicates simulated for 2 tautomers, each docked in 2 initial configurations), we observed sufficient space within the binding site that metformin could reorientate (Supplementary Movie 2). Like thiamine, metformin also formed cation-π interactions with neighboring aromatic residues such as F244 and Y361 (Fig. 4c, d).

Initial docking of metformin inside the inward-facing binding pocket placed this molecule proximal to either E386 or D474 (Fig. 4e). When docked proximal to D474, we observed substrate release for both tautomers within the 500 ns simulation (n = 6). When docked proximal to E386, both tautomers diffused towards the D474 site (n = 6, Supplementary Table 6, Supplementary Movie 3, 4). Taken together these results show metformin release is possible from either site. Upon examination of E386 and D474, both residues were engaged in salt-bridge interactions with R439 and K214, respectively, as well as an ordered hydrogen-bonding network with surrounding residues (Fig. 4f). It should be noted that in the outward-open structures of other OCTs, E386 was solvent-accessible and not engaged with R439 and could represent a key difference in the outward- and inward-open recognition of metformin[27,40].

### OCT1 coordination of fenoterol exhibits stereoselectivity

Fenoterol in OCT1-FNT exhibited coordination in the binding pocket without direct interaction to the acidic residues (E386/D474) of OCT1. One of the two aromatic rings of fenoterol in the cryo-EM density was placed in proximity to Q241 that likely formed a hydrogen bond to its phenolic oxygens (Fig. 5a). The density for the other aromatic ring was

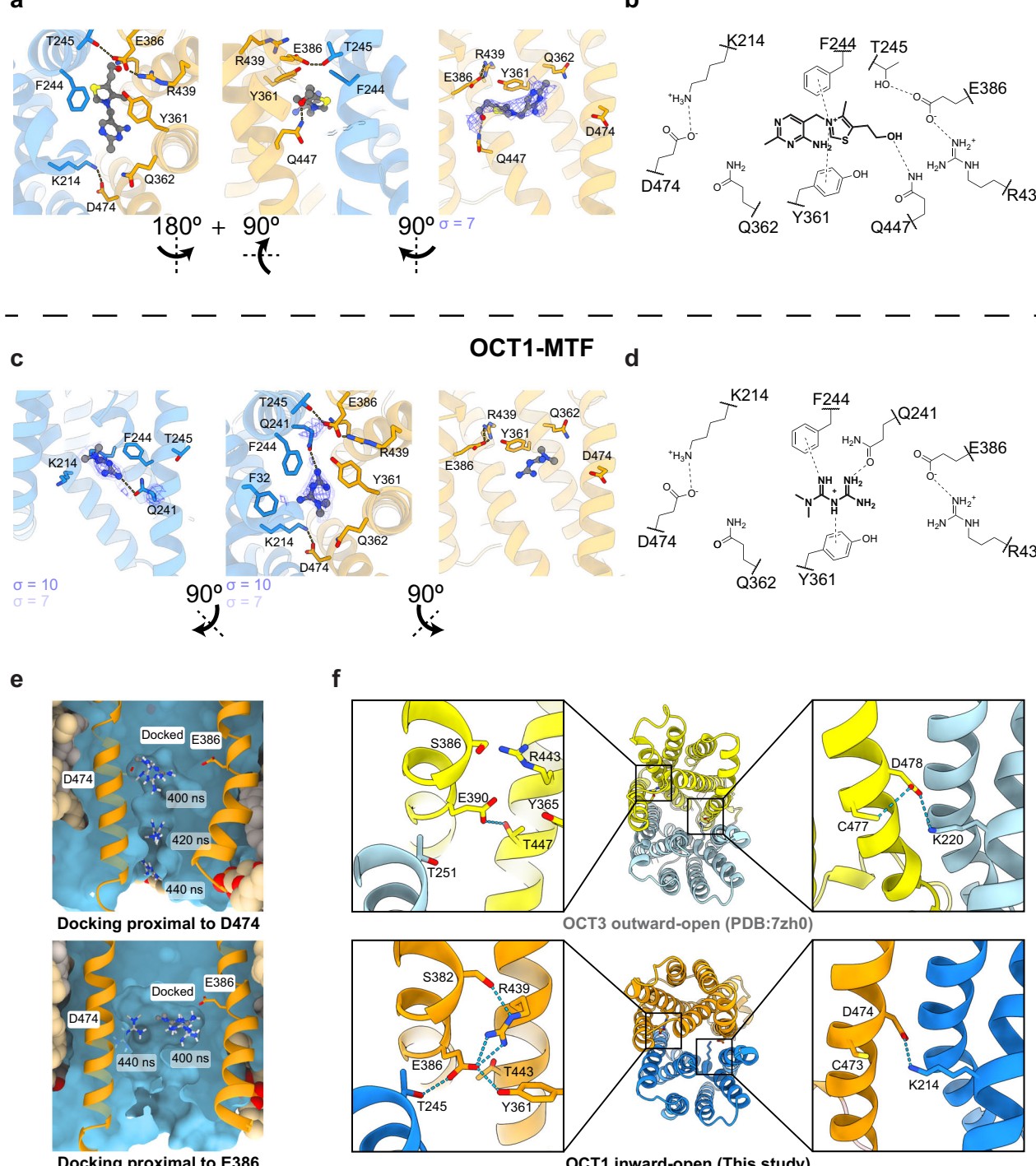

**Fig. 4 | The binding modes of thiamine and metformin to Organic Cation Transporter 1 involve cation-π and polar interactions. a**, **b** Thiamine bound to inward-open OCT1 through cation-π interactions (between the thiazolium ring and F244 and Y361) and a polar interaction between the terminal alcohol and Q447. **c**, **d** Metformin bound to residues Q241, F244 and Y362 in inward-open OCT1, and was distal to E386 and D474 which form salt-bridges to neighboring residues. **e** Snapshots of simulations of metformin initially docked (darker ball representation) proximal to E386 or D474. Position of metformin after 400+ ns showed the molecule diffusing away from the initial site (lighter stick representations). **f** Comparison of key acidic residues in outward OCT3 (PDB:7zh0[27], lighter colors) and inward OCT1 (OCT1-apo, darker colors). The polar network around E386$_{OCT1}$ changed between outward and inward conformations, whereas D474$_{OCT1}$ was always engaged with K214$_{OCT1}$. Similar comparison between OCT1$_{CS}$ and OCT1-apo are shown in Supplementary Movie 1.

comparatively weaker, suggesting it has greater conformational variability because it lacks a similar polar coordination. The resorcinol group was modelled towards Q241 because this optimized the geometry for forming hydrogen bonds with both Q241 and Y361 rather than to the phenol moiety (Fig. 5a). In MD simulations of fenoterol in different conformations, we similarly found that the resorcinol group of fenoterol formed a more stable hydrogen bonding network around the E386 pocket compared to the phenolic group ($n = 3 \times 500$ ns

**OCT1-FNT**

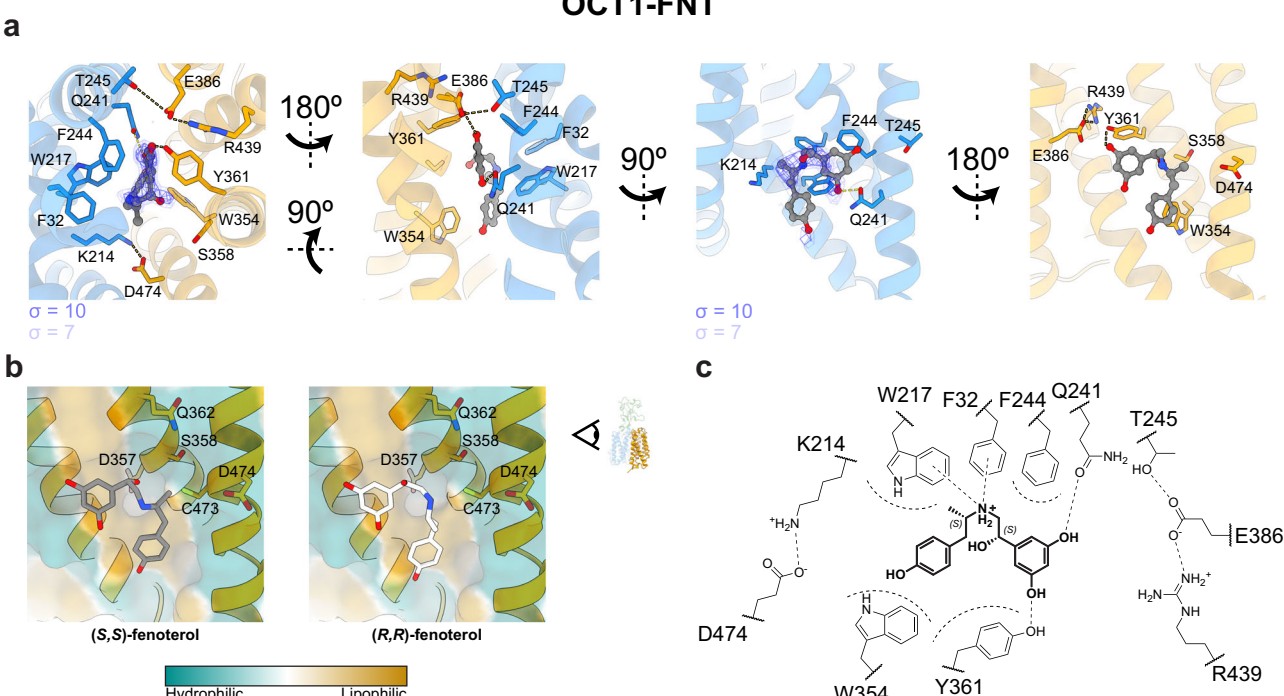

**Fig. 5 | The binding mode of fenoterol to Organic Cation Transporter 1 favors the (S,S)- stereoisomer. a** Fenoterol bound to inward-open OCT1 with hydrogen bonds between the resorcinol group and residues Q241 and Y361. The cryo-EM map contained additional density in the flexible linker of fenoterol that would favor the (S)-hydroxy position over the (R)- isomer. **b** Comparison of the interaction between (S,S)- and (R,R)-fenoterol to the surface of the C-terminal lobe of OCT1. Only the (S)-hydroxy group pointed towards the hydrophilic pocket. **c** Summary of the binding interactions of (S,S)-fenoterol to inward-open OCT1.

simulated for 2 stereoisomers, each docked in 2 configurations, Supplementary Fig. 14, Supplementary Table 9). In the preferred orientation of fenoterol, we observed that while the resorcinol forms stable interactions with the pocket near Q241/E386, the phenol group observes greater flexibility and would rationalize the weaker map density for that region of the molecule. With a pKa of 9.6[43], the majority of fenoterol molecules in the sample and under physiological conditions would be expected to be cationic and protonated at the bridging tertiary amine of the flexible linker in the center of the aromatic pocket.

Differences in substrate uptake between mouse and human homologs of OCT1 are well known and illustrate potential limitations in animal models of drug-OCT1 interactions[39,44]. Fenoterol exhibits higher affinity and lower uptake in human OCT1 compared to mouse OCT1, with studies suggesting that residue 36 (Cys in human and Tyr in mouse) dictates this difference[44]. In our cryo-EM maps, we do not observe direct interaction between C36 and fenoterol, with space above the substrate for the C36Y substitution found in the mouse homolog (Supplementary Fig. 15a). However, in MD simulations, fenoterol sampled conformations that contact K214 and D474, and which is proximal to C36_human (Supplementary Fig. 15b). Hence, the bulky aromatic sidechain of Y36_mouse could form competing interactions with K214 and D474, reducing the affinity of fenoterol to OCT1_mouse as opposed to OCT1_human.

Clinically, fenoterol is used as a racemate consisting of (R,R)- and (S,S)- stereoisomers[45]. Both human OCT1 and OCT2 exhibit different stereospecificity in the transport of this class of compound, with OCT1 favoring transport of (R,R)-fenoterol (by 2-fold) but having a slightly higher affinity for the (S,S)- isomer (higher by 2-fold)[46]. Although we were unable to obtain enantiomerically pure fenoterol for cryo-EM analysis, we observed additional density in the flexible linker of fenoterol that would favor the (S)-hydroxy position over the (R)- isomer (Fig. 5a, Supplementary Fig. 16). The (S)-hydroxyl group did not

directly form polar bonds with any of the nearby sidechains, but points into an open space that is surrounded by polar residues (Fig. 5b, c). However, in the (R)-isomer, the oxygen on the molecule would be oriented so that it would be facing F244, resulting in a less favorable hydrophobic-polar interaction (Supplementary Fig. 16). This subtle difference may explain the similarly small change in the affinity and transport velocity of the two stereoisomers transported through OCT1, where the higher affinity isomer exhibits lower velocity due to more favorable binding interactions. OCT2 on the other hand greatly favors (R,R)-fenoterol in both substrate velocity and affinity[46], and this can be rationalized through the residue differences between OCT1 and OCT2, in particular the steric bulk of Y447_OCT2 hindering the (S,S)-methyl and hydroxy groups (Supplementary Fig. 16).

### The inward-open pocket of OCT1 coordinates diltiazem through charged interactions

We determined the structure of OCT1 bound to diltiazem (OCT1-DTZ) to a resolution of 3.4 Å (Fig. 1g). The OCT1-DTZ structure was solved at pH 6.0 as it was necessary to reduce the pH to dissolve diltiazem (due to its lipophilicity and lower pKa of 7.7[47]), but no significant structural changes were observed when OCT1-DTZ was compared to the other structures solved at pH 8.0.

In OCT1-DTZ, a pendant ammonium group was close to E386 likely forming an electrostatic interaction (Fig. 6). The aromatic groups in OCT1-DTZ were both located near F32, W217, Y361 and I446 which maximized hydrophobic interactions, whereas the additional methoxyphenyl moiety of diltiazem was proximal to W364 (Fig. 6b, d). The interaction of pendant or accessible ammonium moieties to E386 has also been observed in the outward-open structures of OCT1_CS and outward-occluded OCT2_CS, which has been suggested in the mechanism of inhibitor and substrate recognition[40]. Polar interactions between diltiazem and Q241 further stabilize the ligand in the binding site. The hydrophobic pocket of OCT1, with a specific charge coordination

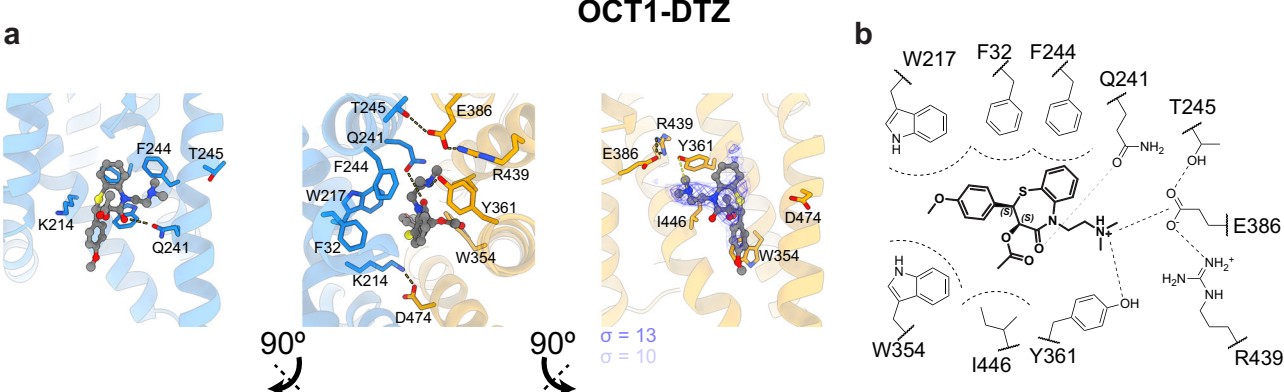

**Fig. 6 | The binding mode of diltiazem to Organic Cation Transporter 1 involves E386. a** Interactions in OCT1-DTZ, with coordination of Q241, Y361, and E386 to the ammonium group of diltiazem. **b** Summary of interactions in the binding pocket of OCT1 with diltiazem.

through E386, likely favors molecules with a pendant ammonium cation tethered to a large lipophilic or aromatic core. Docking followed by MD simulations of diltiazem ($n = 3 \times 500$ ns replicates) correlated well with the modelling of the cryo-EM map (Supplementary Fig. 17, Supplementary Table 10), with an overall RMSD of the ligand of $1.4 \pm 0.4$ Å calculated across the combined 1500 ns of simulation.

## Discussion

Here we present structures of OCT1 in ligand-free and substrate-bound states in the inward-open conformation that show the MFS fold of the protein together with an additional ECD that exhibits limited flexibility. Cationic substrates of OCT1 bind in distinct modes to the inward-open pocket, indicating a range of substrate-protein interactions mediated through aromatic, polar, and charged residues. These structures provide new insights on the mechanism of substrate transport and inhibition of OCT1.

It has been suggested that the ECD in other MFS fold proteins may play a role in gating and oligomerisation[48–50]. In our structures, we observed a hairpin in the ECD that extended over the C-terminal lobe. However, no contacts could be observed between these two domains in the inward-open structure and therefore they are unlikely to be involved in extracellular gating. Previous studies have suggested that the ECD of SLC22 members may also be involved in oligomerization[49,50]. The locations of the disulfides in our structures of inward-open OCT1 suggested that intermolecular disulfides are unlikely to be formed (Fig. 3). Because genetic polymorphisms and mutagenesis of residues in the ECD appear to reduce its function[12,16,50–53] while not appearing to be involved in substrate recognition, we hypothesize that the ECD is instead more likely involved in the folding and trafficking of SLC22 members[31,50,52–55].

In our structures of ligand-bound OCT1, we can divide our panel of substrates into two groups with distinct binding interactions in the inward-open state. In the structures of OCT1 bound to thiamine (OCT1-THA), metformin (OCT1-MTF) and fenoterol (OCT1-FNT), substrates engaged in interactions with predominantly polar and aromatic residues of the inward-open OCT1 binding pocket and did not interact with either of the two critical acidic residues (E386 and D474). On the other hand, the structure of OCT1 in complex with diltiazem (OCT1-DTZ) showed this compound formed electrostatic interactions with E386 through its pendant ammonium cation. E386 is conserved amongst SLC22 members whereas D474 is only conserved in OCTs and is instead arginine in the majority of organic anion transporters (OAT) and novel organic cation transporters (OCTN) in the SLC22 family (Supplementary Fig. 12)[56]. For OCT1-MTF, a possible explanation for the separation of metformin from E386 could be from the spread of the positive charge across conjugated or aromatic systems[57]. The electronegativity

of E386 is partially shielded by neighboring R439 in the inward-facing structure (and less so in outward-facing OCT structures[40]) and hence the relative electrostatic attraction between metformin and E386 is poor. In comparison, the smaller localization of a positive charge in the pendant ammonium of OCT1-DTZ generates a stronger interaction with E386 and these compounds are stabilized by additional hydrophobic interactions. An exception however arises in the case of OCT1-FNT and OCT1-THA, where the positive charge was centralized and/or was not accessible to E386 due to steric hindrance, instead requiring stabilization through cation-π interactions (Figs. 4, 5).

Based on the structural differences in substrate recognition between the outward and inward conformations of OCTs, we propose the following model as a general mechanism for substrate translocation in human OCT1 (Fig. 7). It has been well established that MFS transporters undergo an alternating access transition between outward-facing and inward-facing conformation through an occluded transition state (Fig. 7a)[5,36]. Cationic substrates of OCT1 such as metformin enter the outward-open state and bind to aromatic/hydrophobic residues and/or exposed E386 (Fig. 7c). Closing of the extracellular gate of TM7 captures the substrate, entering an outward-occluded state (Fig. 7a). Opening of the intracellular gate in the inward-occluded state by movement of TM10, TM11, and the ICH bundle opens the cytoplasmic translocation pathway in the inward-open conformation. During the transition from outward-facing to inward-facing (Fig. 7d), R439 re-orients towards E386, neutralizing the charge. This effectively releases substrates with a weaker cationic interaction to E386 to coordinate to Q241/Q447 before exiting the transporter (Fig. 7e). Substrates such as fenoterol and thiamine, that contain a positive charge that is sterically hindered from E386, are instead stabilized by the local aromatic environment and through polar interactions near E386 (Supplementary Fig. 11, 12, Supplementary Tables 7, 9). Comparing the electrostatic potential between a homology model of the outward OCT1 and inward OCT1-apo showed a significant decrease in the electronegativity of the binding pocket between these two conformations (Supplementary Fig. 18) and could also contribute to substrate release as was observed in molecular dynamics simulations of OCT1 and metformin (Fig. 4e, Supplementary Movie 3, 4). Compounds with cationic ammonium pendant groups tethered to an aromatic or hydrophobic core pose significant risks as high-affinity inhibitors of OCT1 as seen in OCT1-DTZ (Supplementary Fig. 1). These compounds can overcome the reduced electronegativity of E386 in the inward-open state and consequently fit tightly to the shape of the binding pocket (Fig. 7f). Larger compounds such as verapamil and imatinib[3,21,40] would also be of concern as inhibitors as their binding interaction in the outward-open state would prevent closure of the extracellular gate due to steric bulk (Fig. 7b)[27,40].

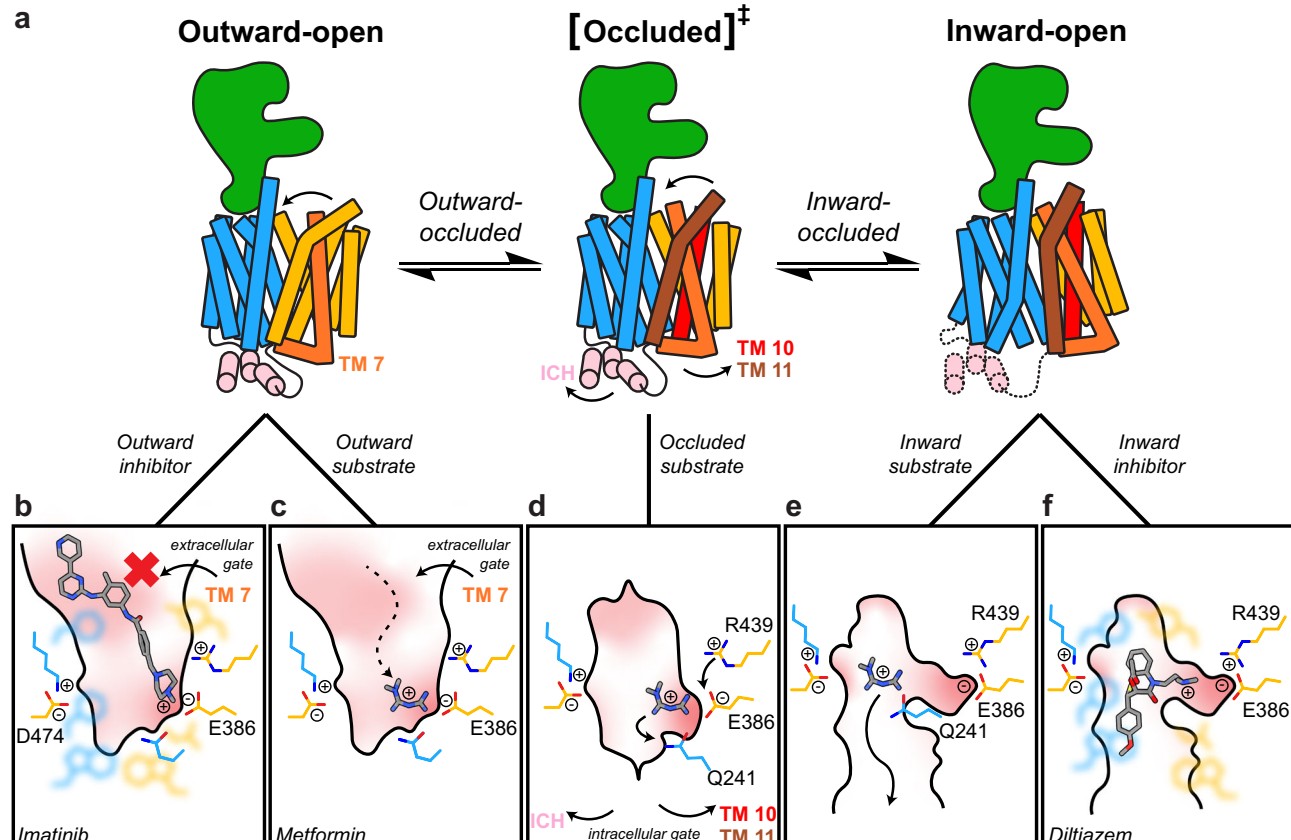

**Fig. 7 | Proposed mechanism of substrate transport and inhibition in Organic Cation Transporters. a** Schematic showing the conformational changes of the C-terminal lobe in the alternating access model. From the outward-open conformation, closing of the extracellular gate of TM7 captures the substrate and causes the transporter to enter the occluded transition state (via an outward-occluded intermediate). Additional rotations of the C-terminal lobe (with the greatest movement for TM10 and TM11) and movement of the ICH opens the intracellular gate to arrive at the inward-open conformation. The ICH in the inward-open conformation is labile in this state. **b–f** Close-up schematic of the binding pocket during the alternating access mechanism and modes of inhibition, with aromatic/hydrophobic residues as faint representations. **b** Large inhibitors of OCT1 (e.g. imatinib) prevent closure of the extracellular gate due to steric hindrance. **c** Substrates of OCT1 (e.g. metformin) bind in the outward-open state to E386 and/ or the hydrophobic pocket. **d** Closure of the extracellular gate rearranges the substrate pocket in the occluded state to neutralize the negative charge of E386 via re-orientation of R439. **e** A decrease in the electronegative potential in the inward-open state frees the substrate from the binding site. **f** High-affinity inhibitors (e.g. diltiazem) extend a pendant ammonium moiety into the electronegative pocket in the inward-conformation and are further stabilized by hydrophobic and π−π interactions.

In summary, this study has identified key molecular interactions between the substrate binding pocket of OCT1 and different classes of compounds and showed the conformational changes required for OCT1 to transition between outward- and inward-facing conformations to facilitate transport. These structures highlight that the inward-open pocket of OCT1 is complex and accommodates different scaffolds in unique ways. Furthermore, our results indicate the importance of identifying the potential interactions of cationic drugs and molecules with OCT1 at different stages of the substrate transport cycle and provides a framework for evaluating the transport of different substrates and the ways in which one may influence the transport of another.

## Methods

### OCT1 protein expression and purification

Full-length human wild-type OCT1 was cloned into the BacMam vector[58] comprising of C-terminal eGFP, 2xFLAG-tag and 8×His-tag preceded by a PreScission cleavage site (OCT1-3C-eGFP-FLAG-His). Baculovirus was generated in MAX Efficiency DH10B cells using the manufacturer's protocol (ThermoFisher) and amplified to P2 in Sf9 cells. HEK293S GnTI[-/-] cells (ATCC) were grown in Freestyle 293 media supplemented with 2% (v/v) FBS. Cells were infected with 5% (v/v) P2 baculovirus once reaching $2.5–3 \times 10^6$ ml$^{-1}$ density. After 24 hours of shaking incubation at 37 °C with 8% $CO_2$, 10 mM sodium butyrate was added to the culture and the incubation temperature was lowered to 30 °C. After 24 h, the cells were harvested by centrifugation at $5500 \times g$. The cell pellets were flash frozen in liquid nitrogen and stored in −80 °C.

A total of 10–15 g cells (from 1 to 1.5 l cultures) were used per preparation and all the purification steps were performed at 4 °C. The cell pellet was resuspended using a syringe in 100 ml of lysis buffer containing 10 mM benzamidine, 10 mM 6-aminocaproic acid, 10 mM DTT, and 1 mM PMSF in 2×TBS (40 mM Tris, pH 8.0, 300 mM NaCl). The cells were lysed by sonication with $2 \times 15$ s pulses (0.9 s on and 0.1 s off). The cell debris was removed by centrifugation at $10,000 \times g$ for 20 min and the membranes pelleted by ultracentrifugation at $100,000 \times g$ for 30 min. The membranes were solubilized in 25 ml of solubilization buffer (2×TBS, 1% LMNG, 0.1% CHS, 10 mM benzamidine, 10 mM 6-Aminocaproic acid, 15 mM imidazole) by gentle stirring for 1 h. The insoluble material was separated by centrifugation at 20,000 g for 20 min. The supernatant was then applied to 2 ml of Ni-NTA agarose (Qiagen) in a gravity column pre-equilibrated in solubilization buffer. The column was washed with 10 column volumes of 2×TBS, 0.1% LMNG, 0.01% CHS, 15 mM imidazole, followed by 10 column volumes of 40 mM Tris, pH 8.0, 300 mM NaCl, 0.1% LMNG, 0.01% CHS, 30 mM imidazole. The OCT1 protein was eluted with 2xTBS, 0.1% LMNG, 0.01% CHS, 150 mM

imidazole. Fractions containing OCT1-3C-eGFP-FLAG-His protein (assessed by SDS PAGE) were pooled and concentrated to 500 µl using a 100 kDa Amicon Ultra-4 (Merck), before application to a Superose 6 10/300 GL size-exclusion column (Cytiva) equilibrated in TBS (20 mM Tris, pH 8.0, 150 mM NaCl). Fractions containing OCT1-3C-eGFP-FLAG-His (assessed by SDS-PAGE) were pooled and 100 µl of 0.8 mg ml$^{-1}$ HRV3C-His protease was added for 1 h to cleave all the tags. Cleaved protein was collected by passing it over 2 ml Ni-NTA beads (pre-equilibrated with TBS) and flow through collected. The beads were then washed with $2 \times 2$ ml TBS. The flow through and the wash fractions were pooled and concentrated to 500 µl and applied to Superose 6 10/300 GL column. The fractions containing OCT1 protein were pooled and concentrated to ~1.0–1.5 mg ml$^{-1}$. For OCT1-DTZ, the second size-exclusion chromatography after cleavage of purification tags was performed in 50 mM citrate, pH 6.0, 150 mM NaCl to obtain the protein at pH 6.0.

## Cryo-EM grid preparation
For the OCT1-apo, 3 µl of purified protein was transferred to a glow-discharged holey gold grid (UltrAuFoil R1.2/1.3, 300 Mesh). Grids were blotted for 4 s at 22 °C, 100% humidity, and flash-frozen in liquid ethane using a Vitrobot Mark IV (ThermoFisher). For OCT1-THA and OCT1-MTF, the ligands were first dissolved in water before adding to a final concentration to 10 mM. For OCT1-DTZ and OCT1-FNT, the ligands were first dissolved in DMSO and added to a final concentration of 1 mM, resulting in a final concentration of 0.5% DMSO in the sample. The protein and ligand mixtures were incubated on ice for 30 min before vitrification. For OCT1-DTZ, UltrAuFoil R0.6/1.0, 300 Mesh grids were used instead.

## Cryo-EM data collection
Grids were transferred to a Thermo Fisher Talos Arctica transmission electron microscope (TEM) operating at 200 kV and screened for ice thickness and particle density. Grids were subsequently transferred to a Thermo Fisher Titan Krios TEM operating at 300 kV, equipped with a Gatan BioQuantum (BioContinuum for OCT1-MTF) energy filter (with 15 eV slit for all ligands except OCT1-MTF where a 20 eV slit was used) and a Gatan K3 Camera. Automatic data collection was performed with EPU 1.2 (SerialEM 4.0 for OCT1-MTF) with a defocus range of −0.5 to −2.0 µm. For OCT1-apo, 7,824 movies were collected at a magnification of 135,000× resulting in a pixel size of 0.86 Å. A total dose of 88 electrons per Å$^2$ spread over 105 frames was used, with an exposure time of 7.0 s. For OCT1-THA, 13,622 movies were collected at a magnification of 135,000× resulting in a pixel size of 0.84 Å. A total dose of 86 electrons per Å$^2$ spread over 110 frames was used, with an exposure time of 7.3 s. For OCT1-MTF, 7728 movies were collected at 81,000× in super resolution mode with a super-resolution pixel size of 0.528 Å. A total dose of 50 electrons per Å$^2$ was used and spread over 50 frames, with an exposure time of 2.9 s. For OCT1-FNT, 15,678 movies were collected at a magnification of 135,000× with a pixel size of 0.84 Å. A total dose of 81 electrons per Å$^2$ was used and spread over 100 frames, with an exposure time of 5.3 s. For OCT1-DTZ, 18,499 movies were collected at a magnification of 135,000× yielding a pixel size of 0.84 Å. A total dose of 79 electrons per Å$^2$ was used and spread over 100 frames, with an exposure time of 6.7 s.

## Cryo-EM data processing
All image processing and reconstruction was performed in cryoSPARC v4.2.1[59,60]. The processing workflow for each of the cryo-EM structures are summarized in Supplementary Figs. 2–6. Movies were pre-processed through Patch Motion Correction and Patch CTF Estimation. Particle picking was performed using the blob picker or template picker (after blob picking on a subset of micrographs) followed by particle extraction with a box size of 256 pixels and Fourier-cropped to 128 pixels. The particles were then subjected to a series of 2D

classifications to remove "junk" particles as well as to sort different orientations of OCT1. The selected particles were used for reference-free Ab-initio Reconstruction (2 or 3 classes, Max resolution = 6 Å) to generate initial maps and classify the particles for 3D refinement. For OCT1-FNT and OCT1-DTZ, Ab-initio maps were used as inputs for Heterogenous Refinement (2 or 3 classes; a lowpass-filtered map of OCT1-apo was added as an additional class) to further classify the particles in these datasets. The particles from the best class from Ab-initio or Heterogenous Refinement were then re-extracted to 256 pixels without down-sampling and refined using Non-Uniform Refinement (Initial lowpass resolution = 12 Å). Local Refinement were performed using a mask of the protein density (encompassing the transmembrane and ECD) to enhance the protein density inside the detergent micelle. For OCT1-MTZ, the particles were initially extracted at a box size of 480 pixels and Fourier cropped to 120 pixels for classifications. For 3D refinement, OCT1-MTZ particles were re-extracted to a box size of 414 pixels and Fourier-cropped to 256 pixels to achieve a final pixel size of 0.854 Å/pix. Data collection and refinement statistics are provided in Supplementary Table 1.

For OCT1-apo, additional focused Local Refinement using a soft mask of only the transmembrane region of the map was performed, followed by 3D classification (5 classes with Force hard classification) and 3D variability analysis (3 components) in cryoSPARC to assess local flexibility of the ECD (Supplementary Fig. 11).

## Model building
Models were built into the cryoSPARC sharpened maps and refined in Coot v0.9[61], PHENIX v1.20.1[62] and ISOLDE v1.5[63]. The AlphaFold[64] model for human OCT1 was used as a guide for the OCT1-apo and the OCT1-apo model was used as a guide for all other structures. The Swiss-Model server[65] was used to generate an outward-open homology model of OCT1 in the outward-open state using the structure of OCT3[27] (PDB: 7zh0). Supplementary Table 2 contains information for model refinement and validation statistics.

Cryo-EM data collection was performed with SerialEM v4.0 and EPU v1.2. Data analysis was performed with cryoSPARC v4.2.1, Coot v0.9, PHENIX v1.20.1, ISOLDE v1.5, Prism v9.5.1, VMD v1.9.4a55, GROMACS 2021.4 MD package, GROMOS 54a7 forcefield, Automated Topology Builder 3.0, AutoDock Vina 4.2.

## Molecular dynamics simulations
Molecular dynamics simulations of membrane-embedded OCT1 were initiated from the OCT1-apo model using the GROMACS 2021.4 MD package[66] in conjunction with the GROMOS 54a7 forcefield[67]. The GROMOS 54A7 force field was chosen as it has been specifically para-meterised to reproduce the experimental solvation free enthalpy and partition coefficients between polar and nonpolar environments for a range of chemical compounds and is ideally suited for lipid and membrane protein simulations. As the cryo-EM model contained a discontinuity between P283 and P331, the N-terminus and residue P331 were capped with NH$_2$, while the C-terminus and residue P283 were acetylated to avoid the introduction of inappropriate charges. The three disulfide bonds (C50-C121, C62-C102 and C88-C142) were explicitly included, however glycans were not included. The protonation state of the side chains of ionizable residue was assigned according to the pK$_a$ predicted using PROpKa version 2.0[68]. All ionizable residues had pK$_a$ values close to the canonical value, except for D357, which had an elevated pK$_a$ of 6.36. In the cryo-EM structure, the sidechain of D357 is buried inside the C-terminal lobe of OCT1, and D357 was modelled as protonated aspartic acid.

Parameters for metformin (Metf(+)A tautomer ATB ID: 29969, Metf(+)B tautomer ATB ID 36158), thiamine (aminopyrymidyl tauto-mer ATB ID 1245669, iminopyrymidyl tautomer ATB ID: 1246704) and fenoterol ((R,R)-enantiomer ATB ID: 1252116, (S,S)-enantiomer ATB ID: 1253118) at pH 8.0, and diltiazem (ATB ID: 1253541) at pH 6.0 and for

each stereochemistry/tautomerization state were generated using the Automated Topology Builder 3.0[69]. Additional proper dihedrals were introduced to enforce planarity of the metformin biguanidinium system ($k_\Phi = 33.5\ \mathrm{kJ\ mol^{l}\ deg^{-1}}$)[70], and to limit excessive out of plane motion of thiamine pyrimidine substituents ($k_\Phi = 41.8\ \mathrm{kJ\ mol^{l}\ deg^{-1}}$). Rationale for parameterization of metformin and thiamine tautomers can be found in Supplementary Text 1.

A summary of all systems simulated is provided in Supplementary Table 3. In all simulations, the cryo-EM structure of apo human OCT1 was embedded in an 80% POPC and 20% cholesterol lipid bilayer created using the MemGen webserver[71]. Ligands were docked into the relevant ligand cryo-EM density map using AutoDock Vina 4.2[72,73]. Each system was solvated with simple point charge (SPC) water[74], chosen for compatibility with the GROMOS 54A7 forcefield, and 150 mM NaCl for physiological relevance. Counterions were added to ensure overall charge neutrality. All simulations were carried out under periodic boundary conditions. Systems were energy minimized using the steepest descent algorithm then equilibrated with decreasing harmonic restraints on the protein in five sequential 1 ns simulations ($1000\ \mathrm{kJ\ mol^{-1}\ nm^{-2}}$, $500\ \mathrm{kJ\ mol^{-1}\ nm^{-2}}$, $100\ \mathrm{kJ\ mol^{-1}\ nm^{-2}}$, $50\ \mathrm{kJ\ mol^{-1}\ nm^{-2}}$, $10\ \mathrm{kJ\ mol^{-1}\ nm^{-2}}$).

The coordinates from the final frame of each equilibrated system were used as the starting configuration for 500 ns production simulations. To increase the statistical sampling of the OCT1/ligand conformation space and ensure convergence of the docked compounds, non-biased production simulations of each OCT1 system were performed in triplicate and a new random starting velocity was assigned at the start of each replicate simulation. The temperature was maintained at 300 K using the Bussi-Donadio-Parrinello velocity rescale thermostat and a coupling constant of 0.1 ps. The pressure was maintained at 1 bar using semi-isotropic pressure coupling with the Parrinello-Rahman barostat using a 5 ps coupling constant and an isothermal compressibility of $4.5 \times 10^{-5}\ \mathrm{bar^{-1}}$. SETTLE was used to constrain the geometry of water molecules and LINCS was used to constrain the covalent bond lengths of the solute. The electrostatic interactions were calculated using the Particle Mesh Ewald summation, and non-covalent interactions were determined via the Verlet scheme with a 1.0 nm cut-off.

Simulations were visualized using VMD v1.9.4a55[75]. Analysis was performed on frames collected at 1 ns intervals, using MDAnalysis v2.2.0[76] and in-house scripts. Details of specific analyses can be found in Supplementary Tables 6–10. The protein, TM domain, and ligand RMSD for each simulation is provided in Supplementary Fig. 19.

### Generation of stable OCT1-FLAG Flp-In HEK293 cell lines
Human OCT1 with a C-terminal 2xFLAG tag (OCT1-FLAG) was subcloned into pcDNA5/FRT/TO vector to generate inducible stable Flp-In™ T-Rex™ HEK293 cell lines (ThermoFisher). Flp-In HEK293 cells were maintained in B-DMEM (Dulbecco's Modified Eagle Medium supplemented with 10% FBS and 10 µg/mL blasticidin) under 5% CO$_2$ at 37 °C. Prior to transfection, $4 \times 10^6$ cells were seeded into 6-well plates and incubated overnight. A mixture of pcDNA5/FRT/TO/OCT1-FLAG vector (100 ng), pOG44 FLP-recombinase expression vector[77] (500 ng), and P3000 reagent (2.3 µL, ThermoFisher) in 70 µL of Opti-MEM (ThermoFisher) were mixed and pre-incubated with Lipofectamine 3000 (ThermoFisher) at room temperature for 15 min. The resulting transfection mixture was added to the Flp-In HEK293 cells 6-well plates containing B-DMEM. Stably integrated OCT1-FLAG was selected by maintaining cells in BH-DMEM (Dulbecco's Modified Eagle Medium supplemented with 10% FBS, 10 µg/mL blasticidin and 200 µg/ml hygromycin B) for 2 weeks. The stably transfected cells were further propagated prior to use in assays.

### Competitive uptake inhibition assay
OCT1-FLAG stable HEK293 cells were seeded at 30,000 cells per well in B-DMEM and induced overexpression with 200 ng ml$^{-1}$ doxycycline in a 96-well plate and incubated at 37 °C, 5% CO$_2$, for 48 h. Ligands were prepared in water (thiamine, metformin) or DMSO (fenoterol, diltiazem) and diluted in HBSS+ (Hank's buffered saline solution supplemented with 10 mM HEPES, pH 7.4) to the desired concentration. The cells were washed twice with 100 µl HBSS+ and then incubated with 100 µl of ligands at room temperature for 20 min before adding 100 µl ASP+ (4-[4-(dimethylamino)styryl]−1-methylpyridinium iodide) to a final concentration of 2 mM in HBSS for 10 min. Fluorescence were measured at excitation/emission wavelengths of 480/612 nm after exactly 2 min on a FlexStation3 (Molecular Devices). The assay was performed in triplicates with $n = 6$ biological replicates. The measured fluorescence was normalized as a fold change over the negative control (doxycycline-uninduced OCT1-FLAG HEK293 cells) and concentration-response curves were fitted in Prism 9.5.1 (GraphPad Software) using a non-linear regression with variable slope.

### Statistics and reproducibility
The cryo-EM analyses were performed on single protein preparations. Detailed statistics for the sample size, data collection and analysis of cryo-EM data are provided in Supplementary Table 2, and Fourier shell correlation curves are provided in Supplementary Figs. 2–6. Competitive uptake inhibition assays were performed to $n = 6$ biologically independent replicates.

### Reporting summary
Further information on research design is available in the Nature Portfolio Reporting Summary linked to this article.

## Data availability
The data that support this study are available from the corresponding authors upon request. The cryo-EM maps have been deposited in the Electron Microscopy Data Bank (EMDB) under accession codes EMD-40334 (OCT1-apo); EMD-40339 (OCT1-THA); EMD-40337 (OCT1-MTF); EMD-40336 (OCT1-FNT); and EMD-40335 (OCT1-DTZ). The atomic coordinates have been deposited in the Protein Data Bank (PDB) under accession codes 8SC1 (OCT1-apo); 8SC6 (OCT1-THA); 8SC4 (OCT1-MTF); 8SC3 (OCT1-FNT); and 8SC2 (OCT1-DTZ). Molecular dynamics simulation trajectories are available at https://github.com/OMaraLab/OCT1_2023 [https://doi.org/10.5281/zenodo.8361638]. Data points for Supplementary Fig. 1 (competitive inhibition assay) are provided in Source Data 1, and original uncropped SDS-PAGE gel in Supplementary Fig. 2a is provided in Source Data 2. Atomic models of OCT1$_{CS}$ and OCT3 can be accessed under accession codes 8ET6 and 7ZH0 respectively. Source data are provided with this paper.

## Code availability
Molecular dynamics simulation trajectories are available at Github [https://github.com/OMaraLab/OCT1_2023] and also Zenodo [https://doi.org/10.5281/zenodo.8361638].

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

## Acknowledgements

We wish to thank and acknowledge Dr. Marcelo De Farias (Pacific Northwest Center for Cryo-EM) for aiding in data collection. A.G.S. was supported by a National Health and Medical Research Council Investigator Grant APP2016308. R.M.R is the recipient of an Australian Research Council Future Fellowship (FT220100717) funded by the Australian Government. We acknowledge the use of the Victor Chang Innovation Centre, funded by the NSW Government, and the Electron Microscope Unit at UNSW Sydney, funded in part by the NSW Government. We also acknowledge the use of the University of Wollongong Cryogenic Electron Microscopy Facility at Molecular Horizons. A portion of this research was supported by NIH grant U24GM129547 and performed at the Pacific Northwest Centre for Cryo-EM at OHSU and accessed through EMSL (grid.436923.9), a DOE Office of Science User Facility sponsored by the Office of Biological and Environmental Research. Molecular dynamics simulations were supported by resources and services provided by the National Computational Infrastructure (NCI), which is supported by the Australian Government; and the University of Queensland's Research Computing Centre (RCC). This research was conducted by the Australian Research Council Industrial Transformation Training Centre for Cryo-Electron Microscopy of Membrane Proteins for Drug Discovery (IC200100052).

## Author contributions

Y.C.Z. and A.G.S. conceived the study. Y.C.Z performed cloning, protein expression and purification. Y.C.Z., M.S. and S.H.J.B. performed the cryo-EM analysis. A.Q. performed and analyzed the molecular dynamics simulations. Y.C.Z. and N.J.S performed the uptake inhibition assays. J.I.V., R.R., M.L.O. and A.G.S. supervised the study.

## Competing interests

The authors declare no competing interests.
