## [Peer Review File · Nature Communications]

Structural Basis of Promiscuous Substrate Transport in Organic Cation Transporter 1Reviewers' Comments:

Reviewer #1:

Remarks to the Author:

The manuscript by Zeng et al., "Structural Basis of Promiscuous Substrate Transport in Organic Cation Transporter 1", describes a set of cryo-EM structures of organic cation transporter 1 (OCT1) in an inward facing state, in apo-form and in complexes with various drugs known to act as OCT1 substrates. The authors provide a detailed description of OCT1-drug interactions.

This is a great study which I like very much. I think the authors' interpretations have been robust, but I would like to see a careful comparison of the apo- and drug-bound substrate pocket densities. There is clearly something observed in the apo-state of OCT1, and the reader would need reassurance that the weakly bound substrates, even at 1-10 mM concentrations used in this study, indeed are represented by the observed densities. This is the main point I have (points 1-2 below), but I also have a few minor comments that I might help the authors to better present their findings for the readers.

1. Fig. 1c, d-h. The density in the substrate pocket in 1c (apo state) would be very good to explore more and to compare in greater detail with the other structures in the supplementary, showing the density under several different contour levels. Is this Tris? Something else? The authors opted not to dwell on this, but I think it is a critical piece of the story, as apo-state is the one to which all drug-bound states are compared.
2. Related to this, Suppl. Fig 3, bottom. The densities of the drugs are great to see (although they are not perfectly matching the compounds in some cases, possibly related to the chosen contour levels). But it would be nice to see a side by side comparison with the apo-state of OCT1 (see point 1 above), ideally contoured at several different levels so that the reader can appreciate that the density observed is indeed drug-related, and not something else present in the apo protein's binding site. Considering that drug-bound states are the central point of the study, it makes sense to carefully and systematically present the data in comparison with the cryo-EM map of the apo-state - this will make the study stronger. I think it is clear that something can bind in the pocket, which is ok - the authors need to present a thorough comparison of the maps, on which they base their conclusions. This is important especially considering that the affinities of the compounds for the protein are so low. The authors use very high concentrations of the compounds, and the risk of over interpreting the densities that correspond to the weak binders is very high.
3. Suppl. Fig. 1 and Table 2. Was statistical comparison performed for the data in the curves or for the IC50 values? If yes, please include.
4. Line 49 - a reference might be good to add here.
5. Fig. 2e. It would be interesting to see a comparison of the gate residues in an outward-open state, either in OCT3 or in the OCT1 homology model.
6. Fig. 3 - is this figure really useful as a main figure? It seems like a relatively low content figure that could be rather sent to the supplementary information.
7. Would the blurry hydrophobic residues be needed in the Fig.7? It may be worth trying to see what the visual effect is of removing them. Essentially they do not provide much useful information, but they add complexity to the figure that may distract the reader making it more difficult to convey the important and specific information about the key residues.
8. Fig. 7f legend - "High-affinity competitive inhibitors." - an example could be good to mention here, similar to imatinib and metformin references earlier in the legend.

9. Suppl. Fig. 2C and others showing similar aspects of the work - the elements shown in the figure panels are very small. The fonts are tiny. For those wishing to print this out, it may be a challenge to see what is shown. Inclusion of panels of this quality feels perfunctory - if these panels are to be included, they should be enlarged sufficiently to see them well.

Reviewer #2:

Remarks to the Author:

The manuscript of Zeng et al. is the third to report cryo-EM data of an SLC22 transporter within the last year. Nevertheless, it may be interesting in several aspects. First, it analyzes OCT1 structure not only in apo and with inhibitors, but also in presence of key substrates like metformin, fenoterol and thiamine. Second, the structures are in inward-open conformation, which enables comparisons with the previously published structures (in outward-open or outward occluded) to generate hypotheses on possible mechanisms of transport. Finally, it reports the best resolution of the extracellular loop between TMH1 and TMH2 up to now. Here are my comments on the manuscript:

1)

Please state clearly early in the manuscript (optimally already in the title and/or abstract) that the analyzed structure is the full-length human OCT1. This is an important information as it was not the case in the previous cryo-EM studies and, historically, rat OCT1 was mostly used in structural analyses.

2)

The hypotheses about the potential conformational changes between the outward and inward-facing conformation of OCT1, though highly interesting, are still hypotheses. They have not been supported by molecular dynamics or any other computational or experimental evidences. This should be clearly stated by the authors in the text.

3)

Connected to the previous comment, the study of Egenberger et al. (JBC 2012, PMID: 22810231) should be discussed. Egenberger et al. reported the presence of a hinge domain in TMH11 that is involved in the substrate-induced conformational changes. Is this supported by the current structural data? If yes, how does it relate to the other conformational changes suggested by the authors?

4)

The structural data from this study should be put into the context of the recently published functional data of fenoterol transport. Meyer et al. (JBC 2022, PMID: 35469921) have reported that the amino acid difference C36Y between mouse and human OCT1 is essential and sufficient for changing the affinity for fenoterol from highly affine (human OCT1) to low affine (mouse OCT1). Meyer et al. performed uptake experiments with a series of substrates that pointed to the second benzene ring to be involved in this interaction. Please discuss the potential of Cys36 involvement in fenoterol binding in the resolved structures, or potentially in an outward-open conformation.

5)

Please report the fold change in ASP+ uptake in the OCT1-overexpressing compared with the control cells and give information whether the inhibition at high concentrations led to complete reduction of the uptake level back to those of the control cells.

6)

The authors reported that the thiamine and metformin density was observed only at low threshold, but was sufficient to model orientation. I do have two related questions. First, at what concentration were the ligands used in the experiment and how does it relate to the K_m -s. Second, metformin has a

highly symmetric structure (this is true to a lesser extent for thiamine and fenoterol). How sure is the exact orientation of the molecule? Please comment critically on this in the manuscript.

Minor comments:

1) Abstract

In contrast to other hepatic uptake transporters, almost no therapeutically relevant drug-drug interactions involving OCT1 are known. Therefore, the authors may consider omitting drug-drug interactions from the abstract.

2) Abstract

Substrates of OCT1 are not only drugs with relevance in cardiovascular indications, but also in pain relief and nausea. Please state this clearly in the abstract, or remove the focusing on cardiovascular. The data of this study may have much broader application.

3) Introduction, lines 40-41

Please refer to the original paper describing thiamine as OCT1 substrate – Chen et al. PNAS 2014 (reference 11 in the current manuscript).

4) Figure 2b

The simultaneous visual representation of two TMHs in outward and inward-open conformations is not clear. Please consider improving, e.g. by choosing more clearly distinguishable colors.

5) Results, lines 203-204

In reference 30, the authors discussed two highly different roles of E386 and D474. E386 should be directly involved in the interaction with the positive charge of the ligands, D474 not. Please state this clearly in the text.

6) Results, lines 266-268

Not only inhibitors were used in reference 30. The authors resolved OCT2 structure also with MPP⁺, which is a classical model substrate of OCTs.

7) Discussion, lines 280-282

There is a difference between the two papers referred to. One suggests intra- the other intermolecular disulfide bonds. Please correct your statement.

8) Methods and supplementary

Please refer to the experiments represented in supplementary figure 1 and supplementary table 2 as "inhibition of the uptake of the model OCT-substrate ASP⁺". Using "dye" is not typical for the field and not precise enough.

We thank both reviewers for their helpful and constructive comments and have modified the manuscript along the lines suggested. To address the risk of over-interpretation raised by the reviewers, we have strengthened the manuscript by performing MD studies that support the binding geometries proposed for thiamine, metformin, fenoterol and diltiazem. However, the MD studies of OCT1 and propranolol were inconclusive, and therefore we plan to include our studies of OCT1-propranolol in a future manuscript.

Please find our point-by-point reply to the reviewer comments below. For clarity, reviewers' comments are in black, our replies are in red, and modified text is in *Italic*.

Reviewer #1 comments (structure)

The manuscript by Zeng et al., "Structural Basis of Promiscuous Substrate Transport in Organic Cation Transporter 1", describes a set of cryo-EM structures of organic cation transporter 1 (OCT1) in an inward facing state, in apo-form and in complexes with various drugs known to act as OCT1 substrates. The authors provide a detailed description of OCT1-drug interactions.

This is a great study which I like very much. I think the authors' interpretations have been robust, but I would like to see a careful comparison of the apo- and drug-bound substrate pocket densities. There is clearly something observed in the apo-state of OCT1, and the reader would need reassurance that the weakly bound substrates, even at 1-10 mM concentrations used in this study, indeed are represented by the observed densities. This is the main point I have (points 1-2 below), but I also have a few minor comments that I might help the authors to better present their findings for the readers.

We would like to thank reviewer #1 for their positive and helpful comments and recommendations to improve the manuscript.

1. Fig. 1c, d-h. The density in the substrate pocket in 1c (apo state) would be very good to explore more and to compare in greater detail with the other structures in the supplementary, showing the density under several different contour levels. Is this Tris? Something else? The authors opted not to dwell on this, but I think it is a critical piece of the story, as apo-state is the one to which all drug-bound states are compared.

We were unable to identify the material present in the ligand density of the apo state by LC-MS and other methods. As suggested by the reviewer, in the revised text we hypothesize that this may be the buffer or potentially another small molecule cation that was retained from the cell during purification. The density is weak above a contour level of $\sigma = 7$, below which it begins to assume an elongated shape, suggesting it is weakly bound or possibly an average of multiple different chemical species. To address this in the text, we have added the following statements to line 208:

We did also observe weak cryo-EM density in the binding site of OCT1-apo at low thresholds ($\sigma < 9$, Fig. 1c, Supplementary Fig. 5). Although we were unable to determine the identity of this material, we hypothesize this may be a buffer component or an endogenous cationic ligand, or the average of multiple different chemical species, consistent with the well-established ability of OCT1 to bind a variety of ligands. The general shape and volume of the ligand density in OCT1-apo was distinct from that observed with OCT1-THA or OCT1-MTF, where 10 mM of substrate had been incubated with the protein sample for cryo-EM analysis (the highest concentration we tested that still allowed for high resolution cryo-EM data collection and was above the reported K_m of thiamine and metformin^{11,39}).

We have also added Supp Fig 4 to now show the maps at different contour levels of the binding site of all the structures to highlight the difference in density across the apo and ligand-bound structures.

2. Related to this, Suppl. Fig 3, bottom. The densities of the drugs are great to see (although the are not perfectly matching the compounds in some cases, possibly related to the chosen contour levels). But it would

be nice to see a side by side comparison with the apo-state of OCT1 (see point 1 above), ideally contoured at several different levels so that the reader can appreciate that the density observed is indeed drug-related, and not something else present in the apo protein's binding site. Considering that drug-bound states are the central point of the study, it makes sense to carefully and systematically present the data in comparison with the cryo-EM map of the apo-state - this will make the study stronger. I think it is clear that something can bind in the pocket, which is ok - the authors need to present a thorough comparison of the maps, on which they base their conclusions. This is important especially considering that the affinities of the compounds for the protein are so low. The authors use very high concentrations of the compounds, and the risk of over interpreting the densities that correspond to the weak binders is very high.

We agree with the comments of the reviewer regarding that there is potential risk of over-interpreting the density and so have been cautious when discussing this point. As mentioned in point 1 above, we have added Supp Fig 4 to allow the reader to compare the distinct ligand densities we see across each of the cryo-EM maps we obtained.

We also clarify that our modelled ligand poses are not absolute in line 208 that we have “*modelled the probable orientation of these substrates in the binding pocket*”. We have also performed MD simulations of OCT1 with substrates in different conformations to further strengthen the main findings from our cryo-EM structures, namely the interactions between thiamine, metformin, fenoterol, and diltiazem with specific residues of the binding OCT1 pocket.

3. Suppl. Fig. 1 and Table 2. Was statistical comparison performed for the data in the curves or for the IC50 values? If yes, please include.

No statistical comparison has been performed in this study. To avoid confusion, we have removed “*The standard deviation and number of replicates for each compound in the dye-uptake inhibition assays are listed in Supplementary Table 2.*” from the Statistics and reproducibility statement.

4. Line 49 - a reference might be good to add here.

We have added reference 3 (Koepsell 2020, review of OCTs mentioning these compounds) in line 50 to introduce these compounds.

5. Fig. 2e. It would be interesting to see a comparison of the gate residues in an outward-open state, either in OCT3 or in the OCT1 homology model.

We have updated Fig 2e. to now include comparison of the equivalent residues found in the recent OCT1_{CS} outward-open structure to highlight the residues involved in the hydrophobic and polar gate.

6. Fig. 3 - is this figure really useful as a main figure? It seems like a relatively low content figure that could be rather sent to the supplementary information.

We think that it is still better to include this figure because it is currently the best resolved density for the ECD of OCTs and highlights the presence of the disulfide bridges and glycosylation sites that are important in stabilising this fold. In addition, the ECD is unique to the SLC22 family and, although is not involved in the mechanism of transport, it has been shown to be important in the correct trafficking and potentially in oligomerisation of this protein. However, if absolutely required, we could move this figure to the supplementary section.

7. Would the blurry hydrophobic residues be needed in the Fig.7? It may be worth trying to see what the visual effect is of removing them. Essentially they do not provide much useful information, but they add complexity to the figure that may distract the reader making it more difficult to convey the important and specific information about the key residues.

Upon reflection we agree with the reviewer and have removed the blurry hydrophobic residues from the middle 3 panels of Fig.7 to improve readability, although we have retained them in the two inhibitor panels because it is important to illustrate the factors driving promiscuity and binding affinity for hydrophobic cationic drugs such as diltiazem.

8. Fig. 7f legend - “High-affinity competitive inhibitors..” - an example could be good to mention here, similar to imatinib and metformin references earlier in the legend.

We have added “*High-affinity competitive inhibitors such as diltiazem*” to the legend text.

9. Suppl. Fig. 2C and others showing similar aspects of the work - the elements shown in the figure panels are very small. The fonts are tiny. For those wishing to print this out, it may be a challenge to see what is shown. Inclusion of panels of this quality feels perfunctory - if these panels are to be included, they should be enlarged sufficiently to see them well.

We have amended Supp Fig. 2-4 to improve the visibility of the figure and allow for comparison of the ligand binding site across different maps as was suggested by reviewer #1 previously.

Reviewer #2 comments:

The manuscript of Zeng et al. is the third to report cryo-EM data of an SLC22 transporter within the last year. Nevertheless, it may be interesting in several aspects. First, it analyzes OCT1 structure not only in apo and with inhibitors, but also in presence of key substrates like metformin, fenoterol and thiamine. Second, the structures are in inward-open conformation, which enables comparisons with the previously published structures (in outward-open or outward occluded) to generate hypotheses on possible mechanisms of transport. Finally, it reports the best resolution of the extracellular loop between TMH1 and TMH2 up to now. Here are my comments on the manuscript:

We would like to thank reviewer #2 for their evaluation and noting the key ways in which this work extends previous studies. The reviewer's helpful and constructive comments have helped to improve our manuscript considerably.

1)

Please state clearly early in the manuscript (optimally already in the title and/or abstract) that the analyzed structure is the full-length human OCT1. This is an important information as it was not the case in the previous cryo-EM studies and, historically, rat OCT1 was mostly used in structural analyses.

We have amended the abstract and introduction to emphasise that we are using full-length human WT OCT1.

2)

The hypotheses about the potential conformational changes between the outward and inward-facing conformation of OCT1, though highly interesting, are still hypotheses. They have not been supported by molecular dynamics or any other computational or experimental evidences. This should be clearly stated by the authors in the text.

We now reference the recent preprint by Yee, S. W. et al. 2023 BiorXiv for which their deep scanning mutagenesis data, in particular for the residues mentioned in our study, corroborate that residues in the gates are functionally important as mutations of these residues leads to significant loss-of-function. This has been amended in line 158:

Additional deep scanning mutagenesis studies of human OCT1 has also shown that a number of the residues involved in the extracellular gate observed in our structures are also important in transport, as deleterious substitution (e.g. hydrophobic to polar and vice versa) of these residues results in reduced transport³¹.

3)

Connected to the previous comment, the study of Egenberger et al. (JBC 2012, PMID: 22810231) should be discussed. Egenberger et al. reported the presence of a hinge domain in TMH11 that is involved in the substrate-induced conformational changes. Is this supported by the current structural data? If yes, how does it relate to the other conformational changes suggested by the authors?

We thank the reviewer for bringing our attention to this study. Indeed, we do observe that the hinge domain is important to allow for flexible motion of the two segments of TM11, and this is supported in our structural analysis.

We have edited Supplementary Fig. 6b to include the glycines involved in the hinge as well as the following in line 117 to put the Egenberger et al. work in the context of our structures:

It has previously been demonstrated in rat OCT1 that mutation of G478rat (G477human) to serine or cysteine reduces the flexibility of TM11 to be able to transition between inward and outward conformations²⁹. In comparison of OCT structures, the glycine-proline motif of TM11 appears to act as a hinge allowing the upper

segment of TM11 to undergo conformational shifts to fully open the binding site in the outward-open state as well as to then occluded the extracellular gate when changing to inward-open (Supplementary Fig. 6b).

4)

The structural data from this study should be put into the context of the recently published functional data of fenoterol transport. Meyer et al. (JBC 2022, PMID: 35469921) have reported that the amino acid difference C36Y between mouse and human OCT1 is essential and sufficient for changing the affinity for fenoterol from highly affine (human OCT1) to low affine (mouse OCT1). Meyer et al. performed uptake experiments with a series of substrates that pointed to the second benzene ring to be involved in this interaction. Please discuss the potential of Cys36 involvement in fenoterol binding in the resolved structures, or potentially in an outward-open conformation.

We have added the following paragraph at line 297-305 discussing this:

Differences in substrate uptake between mouse and human homologs of OCT1 are well known and illustrate potential limitations in animal models of drug-OCT1 interactions^{39,44}. Fenoterol exhibits higher affinity and lower uptake in human OCT1 compared to mouse OCT1, with studies suggesting that residue 36 (Cys in human and Tyr in mouse) dictates this difference⁴⁴. In our cryo-EM maps, we do not observe direct interaction between C36 and fenoterol, with space above the substrate for the C36Y substitution found in the mouse homolog (Supplementary Fig. 13a). However, in MD simulations, fenoterol samples conformations that contact K214 and D474, and which is proximal to C36_{human} (Supplementary Fig. 13b). Hence, the bulky aromatic sidechain of Y36_{mouse} could form competing interactions with K214 and D474, reducing the affinity of fenoterol to OCT1_{human} as opposed to OCT1_{mouse}.

5)

Please report the fold change in ASP+ uptake in the OCT1-overexpressing compared with the control cells and give information whether the inhibition at high concentrations led to complete reduction of the uptake level back to those of the control cells.

We have updated Supp Fig 1 now as a fold change over the control cells to allow comparison of OCT1-overexpressing and control as well as giving information regarding if complete reduction is observed at high inhibitor concentration

6)

The authors reported that the thiamine and metformin density was observed only at low threshold, but was sufficient to model orientation. I do have two related questions. First, at what concentration were the ligands used in the experiment and how does it relate to the Km-s. Second, metformin has a highly symmetric structure (this is true to a lesser extend for thiamine and fenoterol). How sure is the exact orientation of the molecule? Please comment critically on this in the manuscript.

We agree with the reviewer's comment that we cannot be absolutely sure of the orientation of the molecule. We have addressed the reviewer's concerns and clarified this point in the paragraph line 203-216:

The cryo-EM structures of OCT1-THA and OCT1-MTF in the inward conformation strongly resembled OCT1-apo (Fig. 1, Supplementary Fig. 2, 3). The density corresponding to thiamine and metformin could only be observed at a relatively lower threshold compared to the sidechain density ($\sigma < 15$ for sidechain, $\sigma < 9$ for ligand), likely owing to its low affinity for the transporter (Fig. 4a, c, Supplementary Fig. 3, 5). However, the general contour of the ligands in the map could be established clearly and therefore we were able to model the probable orientation of these substrates in the binding pocket. We did also observe weak cryo-EM density in the binding site of OCT1-apo at low thresholds ($\sigma < 9$, Fig. 1c, Supplementary Fig. 5). Although we were unable to determine the identity of this material, we hypothesize this may be a buffer component or an endogenous cationic ligand, or the average of multiple different chemical species, consistent with the well-established ability of OCT1 to bind a variety of ligands. The general shape and volume of the ligand density in OCT1-apo was

distinct from that observed with OCT1-THA or OCT1-MTF, where 10 mM of substrate had been incubated with the protein sample for cryo-EM analysis (which was the highest concentration we tested above the reported K_m of thiamine and metformin^{11,39} that still allowed for high resolution cryo-EM data collection).

However, to corroborate the general binding poses that we have modelled, we have also performed MD studies on these compounds to corroborate the orientations we observe in cryo-EM. We note that metformin could potentially flip inside the binding pocket due to its small size, as mentioned in line 254:

The density corresponding to metformin appears to be coordinated by Q241 and so the compound was modelled with the desmethylated nitrogen forming putative polar interactions to this residue (Fig. 4c), however in MD simulations of metformin and OCT1 (500 ns, 2 tautomers in 2 initial configurations with $n = 3$ replicates), we observed sufficient space within the binding site that metformin could reorientate (Supplementary Movie 2).

Minor comments:

1) Abstract

In contrast to other hepatic uptake transporters, almost no therapeutically relevant drug-drug interactions involving OCT1 are known. Therefore, the authors may consider omitting drug-drug interactions from the abstract.

We have removed this statement from the abstract.

2) Abstract

Substrates of OCT1 are not only drugs with relevance in cardiovascular indications, but also in pain relief and nausea. Please state this clearly in the abstract, or remove the focusing on cardiovascular. The data of this study may have much broader application.

We have broadened the scope of OCT1-related substrates and inhibitors in the abstract in line 21: *Genetic variations can alter the efficacy and safety of compounds transported by OCT1, such as those used for cardiovascular, oncological, and psychological indications.*

3) Introduction, lines 40-41

Please refer to the original paper describing thiamine as OCT1 substrate – Chen et al. PNAS 2014 (reference 11 in the current manuscript).

We thank the reviewer for picking up this omission and have added the reference to Chen et al. 2014 to the statement in line 44.

4) Figure 2b

The simultaneous visual representation of two TMHs in outward and inward-open conformations is not clear. Please consider improving, e.g. by choosing more clearly distinguishable colors.

We appreciate the reviewer's suggestion to improve the visibility of the two conformations. We have decided to not alter the colour scheme in order to keep this consistent throughout the text to enable the reader to easily identify the domains of the transporter (yellow/blue/green) and the outward vs inward conformations (light and dark). We have experimented with different ways of visualising the helices using other representations (e.g. cylinders, tubes, C-alphas) but were unable to find a satisfactory replacement for the ribbon.

We have added the following text in the Figure 2 legend (*"Detailed comparison of each of the C-terminal lobe helices can be found in Supplementary Fig. 6."*) to direct readers to another representation in the Supp. Figs. to help address this request of the reviewer.

5) Results, lines 203-204

In reference 30, the authors discussed two highly different roles of E386 and D474. E386 should be directly

involved in the interaction with the positive charge of the ligands, D474 not. Please state this clearly in the text.

We have added the distinction between E386 and D474 noted by Suo et al. in the text at line 234:

E386 in the outward-open conformation is solvent-accessible and directly coordinates cations in other OCTs, while D474 appears to be important in structural stability due to its consistent coordination to K214 and may influence the electrostatics of the binding pocket (Fig. 4f)^{40,41}.

6) Results, lines 266-268

Not only inhibitors were used in reference 30. The authors resolved OCT2 structure also with MPP+, which is a classical model substrate of OCTs.

We have amended the text to reflect this in line 338:

The interaction of pendant or accessible ammonium moieties to E386 has also been observed in the outward-open structures of OCT1_{CS} and outward-occluded OCT2_{CS}, which has been suggested in the mechanism of inhibitor and substrate recognition⁴⁰.

7) Discussion, lines 280-282

There is a difference between the two papers referred to. One suggests intra- the other intermolecular disulfide bonds. Please correct your statement.

Upon reflection we have removed mention of the intermolecular/intramolecular cysteines as it does not add new information. This paragraph (line 355-363) has now been updated to:

It has been suggested that the ECD in other MFS fold proteins may play a role in gating and oligomerisation⁴⁸⁻⁵⁰. In our structures, we observed a hairpin in the ECD that extended over the C-terminal lobe. However, no contacts could be observed between these two domains in the inward-open structure and therefore they are unlikely to be involved in extracellular gating. Previous studies have suggested that the ECD of SLC22 members may also be involved in oligomerization^{49,50}. The locations of the disulfides in our structures of inward-open OCT1 suggested that intermolecular disulfides are unlikely to be formed (Fig. 3). Because genetic polymorphisms and mutagenesis of residues in the ECD appear to reduce its function^{12,16,50-53} while not appearing to be involved in substrate recognition, we hypothesize that the ECD is instead more likely involved in the folding and trafficking of SLC22 members^{31,50,52-55}.

8) Methods and supplementary

Please refer to the experiments represented in supplementary figure 1 and supplementary table 2 as “inhibition of the uptake of the model OCT-substrate ASP+”. Using “dye” is not typical for the field and not precise enough.

We have corrected this as suggested to “Potency of compounds on the inhibition of uptake of the model OCT1-substrate ASP+” and removed the “dye” terminology from the paper.

Reviewers' Comments:

Reviewer #1:

Remarks to the Author:

The authors have done a thorough job of addressing the comments, addressing the criticisms in a satisfactory manner. I have no further comments to add.

Reviewer #2:

Remarks to the Author:

The authors addressed all my comments to my highest satisfaction. The manuscript is now substantially improved especially by adding MD to link it to the current inward with the existing outward structures.

Reviewer #3:

Remarks to the Author:

In this manuscript the authors describe a structural and mechanistic study on the Organic Cation Transporter 1 (OCT1). In particular, this work reports new inward-facing structures of OCT1, in the apo state as well as bound to different ligands. Furthermore, the local structural stability, solvation properties and ligand-protein interactions were assessed using a comprehensive set of molecular dynamics simulations.

The work is very well written/presented and the simulation data/analysis are well done from the technical standpoint (setup, sampling and analysis). Simulation statistics/sampling, with repeated independent trajectories, is also sufficient for the purpose of the work and provides a useful complement to the cryo-EM data.

Overall, this work provides an important contribution to the field of membrane transport and there are no relevant changes to be done to further improve it.